# Downstream Effects of Mutations in *SOD1* and *TARDBP* Converge on Gene Expression Impairment in Patient-Derived Motor Neurons

**DOI:** 10.3390/ijms23179652

**Published:** 2022-08-25

**Authors:** Banaja P. Dash, Axel Freischmidt, Jochen H. Weishaupt, Andreas Hermann

**Affiliations:** 1Translational Neurodegeneration Section “Albrecht-Kossel”, Department of Neurology, University Medical Center Rostock, 18147 Rostock, Germany; 2Department of Neurology, Ulm University, 89081 Ulm, Germany; 3Division of Neurodegeneration, Department of Neurology, Mannheim Center for Translational Neurosciences, Medical Faculty Mannheim, Heidelberg University, 68167 Mannheim, Germany; 4Center for Transdisciplinary Neurosciences Rostock, University Medical Center Rostock, 18147 Rostock, Germany; 5Deutsches Zentrum für Neurodegenerative Erkrankungen (DZNE) Rostock/Greifswald, 18147 Rostock, Germany

**Keywords:** amyotrophic lateral sclerosis (ALS), human induced pluripotent stem cells (iPSC), motor neurons (MN), RNA sequencing (RNA-Seq), differentially expressed genes (DEG), protein-protein interaction (PPI)

## Abstract

Amyotrophic Lateral Sclerosis (ALS) is a progressive and fatal neurodegenerative disease marked by death of motor neurons (MNs) present in the spinal cord, brain stem and motor cortex. Despite extensive research, the reason for neurodegeneration is still not understood. To generate novel hypotheses of putative underlying molecular mechanisms, we used human induced pluripotent stem cell (hiPSCs)-derived motor neurons (MNs) from *SOD1*- and *TARDBP* (TDP-43 protein)-mutant-ALS patients and healthy controls to perform high-throughput RNA-sequencing (RNA-Seq). An integrated bioinformatics approach was employed to identify differentially expressed genes (DEGs) and key pathways underlying these familial forms of the disease (fALS). In TDP43-ALS, we found dysregulation of transcripts encoding components of the transcriptional machinery and transcripts involved in splicing regulation were particularly affected. In contrast, less is known about the role of SOD1 in RNA metabolism in motor neurons. Here, we found that many transcripts relevant for mitochondrial function were specifically altered in SOD1-ALS, indicating that transcriptional signatures and expression patterns can vary significantly depending on the causal gene that is mutated. Surprisingly, however, we identified a clear downregulation of genes involved in protein translation in SOD1-ALS suggesting that ALS-causing SOD1 mutations shift cellular RNA abundance profiles to cause neural dysfunction. Altogether, we provided here an extensive profiling of mRNA expression in two ALS models at the cellular level, corroborating the major role of RNA metabolism and gene expression as a common pathomechanism in ALS.

## 1. Introduction

Amyotrophic lateral sclerosis (ALS) is a progressive and irreversible neurodegenerative disease characterized by selective loss of motor neurons in the motor cortex, brain stem and spinal cord. Presently, more than 25 genes have been identified as monogenetic causes of ALS; among these are mutations of the Cu/Zn superoxide dismutase 1 (*SOD1*) and the TAR DNA-binding protein 43 (*TARDBP*) [1,2,3,4]. Despite extensive research, the underlying pathomechanisms in general but also those which are likely causing motor neuron degeneration and cell death in *SOD1* versus *TDP43* mutations remain unknown.

Over the years, studies on various disease mechanisms including endoplasmic reticulum (ER) stress, oxidative stress, excitotoxicity, inhibition of the proteasome, mitochondrial damage, dysregulation of RNA metabolism, axonal disorganization and disrupted axonal transport have been implicated in SOD1- and TDP43-ALS [5,6,7,8,9,10]. In many cases, these diverse downstream abnormal events activate and recruit nonneuronal cells such as astrocytes, microglia, and oligodendrocytes, which trigger and sustain motor neuron degeneration either through the release of neurotoxic or pathogenic factors in response to a wide range of extracellular signals and stress or through the lack of neuronal support (neurotrophic factors) [11,12]. Furthermore, generated functional astrocytes from human pluripotent stem cells (e.g., embryonic stem cells (ESCs) and iPSCs) carrying TDP43 and SOD1 mutations showed that these mutant astrocytes exhibited increased levels of astrocytic toxicity, impaired subcellular localization, decreased cell survival and dysfunction of the neuroprotective response [13,14]. In addition, microarray and high throughput RNA-seq based transcriptomic studies have revealed many cellular defects of motor neuron function in both forms of the ALS models, consistent with the hypothesis that various biological processes, including inflammation, mitochondrial dysfunction, enhanced apoptosis, oxidative damage, protein misfolding, altered axonal transport, RNA metabolism and lipid metabolism contribute to the pathobiology of the disease [15,16,17,18,19]. We recently reported on network-based interactome and transcription factor analysis of iPSC-derived MNs from FUS- and SOD1-ALS patients. The results revealed unique pathways associated with herpes simplex virus infection (*FUS* mutation) and dysregulation of metabolic pathways (*SOD1* mutation) [15].

Among these, alteration in RNA processing mechanisms [1,20] is considered to be one of the most important events and can lead to neural dysfunction and neurodegeneration [21,22,23,24]. Multiple roles in RNA regulation have been identified for *TDP43* such as RNA splicing, translation, transport and microRNA (miRNA) biogenesis [5,25,26,27], suggesting a potential role of altered RNA expression and post translational processing of proteins in the disease. Unlike *TDP43*, *SOD1* does not contain RNA-binding motifs and ALS patients with *SOD1* mutations do not exhibit similar defects in RNA processing [28,29,30,31]. However, reports have demonstrated a potential function of mutant *SOD1* in regulating RNA metabolism including alternative splicing as well [20,32]. A growing body of evidence showed that the axons of mature sensory and peripheral motor neurons strongly rely on mRNA transport and local translation to maintain homeostasis [33,34]. In another finding, the upregulation of ribosome synthesis in axons has been reported in the pathogenesis of both mutant *SOD1*-G93A transgenic mouse models and human ALS autopsy samples [35].

In addition, RNA-seq analyses of the anterior branch of human obturator MNs biopsied from patients with ALS demonstrated upregulation of a cluster of genes that play important roles in biological pathways involving RNA processing and protein metabolism [36]. Accordingly, RNA profiling studies from the axon samples of cultured spinal cord neurons using microfluidic technology revealed that mRNAs and miRNAs are differentially expressed in the somatic compared with the axonal neuronal compartments, showing aberrant axonal RNA metabolism and defects in mitochondrial functions [7,37,38]. Indeed, an elegant study based on Axon-seq investigated transcriptomic changes in mouse embryonic stem cells (mESCs)-derived MNs overexpressing *SOD1*(G93A) showed extensive dysregulation of oxidative energy and ribosome production [39]. Recently, functional enrichment analysis based on single-cell transcriptomic technology also identified several dysregulated pathways related to RNA processing mechanisms including translation, splicing and mitochondrial function in individual neurons obtained from *SOD1* patient-derived iPSCs [40]. Altogether, these transcriptomic investigations support a pathogenic role for dysregulation of RNA processing in SOD1- and TDP43-ALS.

Hence, taking the advantage of iPSC model system and by integrating bioinformatics analysis, we have examined two ALS models, human motor neurons with mutations in *SOD1* and *TDP43*. We addressed whether subtypes of ALS caused by different genetic mutations might be stratified on the basis of transcriptional alterations. Our analysis highlighted distinct clusters of DEGs within both the *SOD1* and the *TDP43* mRNA profiles, supporting previous studies of divergent pathways in different ALS genes. However, further protein-protein interaction (PPI) network analysis revealed also pathways converging on the regulation of protein translation, by alteration of ribosomal protein transcription in *SOD1* mutant and splicing deregulation in *TDP43* mutant motor neurons.

## 2. Results

### 2.1. Motor Neuron Differentiation in iPSC-Derived Cell Lines

We studied iPSC-derived spinal motor neuron cultures from four different non related ALS patients (two of each carrying SOD1 and TDP43, respectively) which were compared to three different healthy control individuals (different families with different mutations; SOD1 D90A, SOD1 R115G, TDP43 S393L and TDP43 G294V, respectively; different controls from different families, for details see also Table 1). Different gene mutations can cause significantly different disease courses and phenotypes. This is especially true in the case of D90A SOD1. Patients carrying D90A SOD1 mutation showed less phenotypic variability than patients with other Cu/Zn SOD1 mutations, with the exception of the variability of the age of onset of first symptoms (patients show the widest span of 74 years) [41]. Cellularly, D90A mutant iPSC-derived motor neurons yield also different mitochondrial and aggregation phenotypes compared to other SOD1 mutations [42]. Since our intention was to search and address for overarching phenotypes we thus included also a D90A cell line.

The generation of human NPCs (neural progenitor cells) and motor neurons was accomplished following the protocol from Reinhardt et al., Bursch et al. and Naumann et al. [43,44,45]. All the iPSC lines we used were low passage number (less than 20) and differentiated for 14–21 days of terminal differentiation (=total DIV 30). All these cell lines have been previously characterized including the acquisition of classical spinal motor neurons markers (described above), electrophysiological function and the sequential appearance of progressive neurodegeneration. Please refer to the citations for detailed information [42,46,47]. In summary, immunolabeling detected the presence of neuronal (TuJ1 80–90%, MAP2 80–90%) and motor neuron specific markers (SMI32 70–75%) in the cells without significant differences in neuron morphologies between wildtype controls (*n* = 3 subjects, 1 clone each) and ALS mutants (SOD1 *n* = 2 subjects, 1 clone each; TDP43 *n* = 2 subjects, 2 clones of one subject and 1 clone of second subject). There was no difference between wildtype and mutants SOD1 [42,46] and TDP43 [47].

### 2.2. RNA-Seq Profiling and Identification of DEGs between SOD1- and TDP43-ALS

A detailed outline of the study protocol is summarized in Figure 1. Our analysis focused on RNA sequencing and bioinformatics analysis of *SOD1*- and *TDP43*-ALS mutant patient-derived spinal motor neurons, as compared to healthy controls (for details of the cell lines see Table 1).

A total of 1448 DEGs were found in SOD1-ALS, with 668 upregulated and 780 downregulated genes. In TDP43-ALS, 1160 DEGs were found, including 626 upregulated and 534 downregulated genes. The cut-off criteria were set at *p*-value ≤ 0.05 and │log_2_FC**│** ≥ 1.5 (FC, fold change). A full list of DEGs is provided in Appendix A.

### 2.3. Functional and Pathway Enrichment Aanalysis of DEGs

Gene Ontology (GO) and pathway enrichment analysis of DEGs were performed by different databases, respectively, using *p*-value ≤ 0.05, and │log_2_FC│ ≥ 1.5 as cut-off value, which were useful in identifying important biological functions of a specific gene. In *SOD1* mutant cells, biological processes (BP) linked to DEGs were significantly enriched in cytoplasmic translation, regulation of Wnt signaling pathway and response to oxidative stress processes. Referring to the analysis of Kyoto Encyclopedia of Genes and Genomes (KEGG), Reactome and Wikipathways indiated that the top pathways associated with DEGs were mainly related to ribosome and eukaryotic translation elongation functions (Figure 2A,B). In *TDP43* mutant motor neurons the DEGs were largely involved in regulation of proteolysis, protein export, growth regulation processes, phagocytosis and glycoproteins/glycans signaling cascades (Figure 2C,D). Collectively, the comprehensive enrichment analysis indicated distinct transcriptional signatures associated with SOD1- and TDP43-ALS motor neurons. These findings are in agreement with previous descriptions of quite different transcriptome changes in SOD1 and TDP43-ALS [38,39,40]. Top ten biological functions enriched for SOD1- and TDP43-ALS DEGs by EnrichR and DAVID are presented in Appendix A and the complete list of all GO/pathway terms is given in Appendix A.

### 2.4. PPI Network Construction and Module Analysis

We used the Search Tool for the Retrieval of Interacting Genes database (STRING) (version 11.5 [48]) and Cytoscape software (Institute for Systems Biology, UCSD, San Diego, CA, https://www.cytoscape.org, accessed on 28 June 2022) (version 3.8.2 [49]) to predict protein-protein interactions (PPI) of the detected DEGs in SOD1 and TDP43 datasets which were subsequently analyzed with the threshold (degrees ≥10) using the Molecular Complex Detection (MCODE) tool [50]. In SOD1, with the confidence at high level, total DEGs (877 nodes and 515 edges, PPI enrichment *p* = 6.62 × 10^−10^) were mapped to generate PPI network (Figure 3A). The analysis by MCODE plugin revealed many functional modules in the whole DEGs network. However, only top two modules (Modules 1 and 2) containing five or more than five nodes were identified (Figure 3C and Appendix A). Metascape (Figure 3E) and ClueGO/CluePedia (Figure 3G) [51,52] enrichment analysis of the functional module with the highest MCODE score showed that all the genes from module 1 (*CCDC124*, *EEF1B2*, *EEF1G*, *PLEC*, *RPL10A*, *RPL11*, *RPL13*, *RPL28*, *RPL3*, *RPL32*, *RPL35A*, *RPL38*, *RPL39*, *RPL5*, *RPL6*, *RPL7A*, *RPLP0*, *RPLP1*, *RPS12*, *RPS24*, *RPS6*, *RPS8*, *RPSAP58*, and *TPT1*, *MRPL34*, *MRPL45*, *MRPL49*, *MRPL30*, and *MRPL28*) exhibited downregulation and were significantly enriched in translational elongation, ribosomes and mitochondrial ribosome functions, while the genes in module 2 (*MT*-*ND3*, *MT*-*ND5*, *MT*-*ND6*, *NDUFA12*, *NDUFA13*, *NDUFA2*, *NDUFA9*, *NDUFB10*, *NDUFB9*, *NDUFC1*, *NDUFS3*, *NDUFS4*, *NDUFS6*, *NDUFS7*, and *NDUFS8*) showed upregulation and were involved in NADH dehydrogenase ubiquinone activity and mitochondrial complex I assembly/OXPHOS system (Appendix A). Thus, PPI network analysis highlighted two main pathways out of the multiple pathways affected in SOD1-ALS above described.

Similarly, in TDP43, the overall PPI network of the total DEGs (upregulated and downregulated) containing 655 nodes and 136 edges (PPI enrichment *p* = 0.0626) was surveyed for identification of functional modules in the network (Figure 3B). In addition, MCODE plugin was used to extract functional modules and based on highest score two significant modules were screened out in the whole DEGs of PPI network. One module included a total of 9 genes; *AK6*, *TAF1*, *TAF10*, *TAF11*, *TAF12*, *TAF4*, *TAF5*, *TAF6*, and *TAF7* which were mainly associated with RNA polymerase based general transcription processes (Figure 3D,F,H), whereas the second module consisted of a total of 7 genes *LSM2*, *LCM3*, *LSM4*, *LSM6*, *LSM7*, *LSM8*, and *SART3* and were mainly enriched in RNA splicing and spliceosomal snRNP assembly functions (Appendix A). Of note, both modules were upregulated in case of TDP43. Thus, also in TDP43-ALS, PPI network analysis highlighted two main pathways affected. Appendix A (with gene names) summarize the modules in PPI networks and their enriched functions, respectively. The complete results of all PPI network analysis and the list of hub binding partners are provided in Appendix A.

### 2.5. Identification of Hub Genes and Transcription Factor Regulatory Network Analysis

In SOD1-ALS, the top 10 key nodes of the total DEGs in the PPI network were selected as hub genes according to the scoring of maximum correlation criterion (MCC) by using the cytoHubba plugin [53], which identifies important nodes and modules by topological algorithms. The hub genes of the network analysis were related to ribosomal functions (*RPLP0*, *RPL11*, *RPL5*, *RPL3*, *RPL13*, *RPL10A*, *RPL6*, *RPS24*, *RPS8*, and *RPS6*) (Figure 4A). Notably, these hub genes were mostly downregulated in motor neurons, suggesting that ribosome misregulation is the most relevant change in SOD1-ALS patients.

On the other hand, using the cytoHubba plugin in Cytoscape [53], a total of 10 nodes were identified as hub genes from the overall PPI network of TDP43-ALS motor neurons (*TAF10*, *TAF1*, *TAF12*, *TAF6*, *TAF5*, *TAF11*, *TAF7*, *TAF4*, *AK6*, and *LSM7*) (Figure 4B). As shown in Figure 4, the majority of the hub genes in the network were upregulated DEGs in TDP43-ALS samples and were significantly involved in RNA processing functions such as transcription and splicing.

To identify key regulators of the SOD1 and TDP43-ALS dysregulated genes, a target gene-transcription factor (TF) regulatory network of the identified ten hub genes was constructed and analyzed in NetworkAnalyst. In the SOD1-ALS analysis, a gene-TF network was performed including 68 interaction pairs among 10 genes and 45 TFs (Figure 4C). While *RPS24* was found to be regulated by 9 TFs, *RPL6* was regulated by 8 TFs, *RPL10A* was regulated by 7 TFs, and *RPLP0* was regulated by 6 TFs. In addition, various TFs were found to regulate more than one hub gene, and four TFs (*PPARG*, *GATA2*, *FOXF2*, and FOXC1) were identified with a connectivity degree ≥5 in the gene-TF regulatory network, indicating that these TFs have close interactions with these hub DEGs (Table 1).

Similarly, the gene-TF regulatory network in TDP43-ALS was also constructed including 54 interaction pairs among 9 genes and 39 TFs (Figure 4D). Of these, TAF5 was found to be regulated by 10 TFs, *LSM7* was regulated by 8 TFs, *TAF6* was regulated by 6 TFs, and *TAF11* was regulated by 5 TFs. Based on a connectivity score ≥5, forkhead Box C1 (*FOXC1*) was predicted to regulate seven genes and GATA-binding factor 2 (*GATA2*) was found to regulate 5 genes (Table 2). The complete list of all TFs is given in Appendix A.

## 3. Discussion

In this study, we have performed systematic transcriptome profiling and bioinformatics analyses to identify genes that were differentially expressed (*p*-value ≤ 0.05) in iPSC-derived MNs from SOD1- and TDP43-ALS patients and healthy controls using high-throughput RNA-Seq technology. Firstly, our data revealed that both SOD1-and TDP43-ALS datasets exhibited different signatures, suggesting the molecular pathogenesis of different genetic ALS forms might exhibit varying extents of genotype and phenotype overlap. Secondly, in depth PPI based modeling identified pathogenic pathways, namely translation and transcription, which converge on defects in RNA metabolism in models associated with mutations in *SOD1* and *TDP*43. While defects in RNA processing, transcription and splicing were expected to be common occurrences in RNA binding protein TDP43 and thus in overall protein landscape architecture, it was surprising to note that the most important hub mechanistic pathway in SOD1 also involved translation.

To further refine the GO and KEGG enrichment analysis, we performed a thorough analysis of the contribution of experimentally-derived PPIs for annotation of proteins and generated a functionally arranged module of terms GO/pathway that were differentially regulated with their significant gene interactions based on the *p*-values and kappa statistics. In SOD1, we identified important genes and their functional interactions and how they were associated with the pathogenesis and progression of ALS. Genes most strongly downregulated in motor neurons were found to be key drivers involved in the differential regulation of the protein translational pathway and ribosomal functions (Figure 3C,E,G). Interestingly, in agreement with a previous gene expression profiling study analyzing iPSC-derived MNs as well as biopsied samples from ALS and motor neuropathy patients, our analysis revealed significant overrepresentation of genes and pathways relevant to ribosome biogenesis and RNA processing [35,36,39,40,54]. Similarly, the ribosomal protein S6 has been observed in the axons of embryonic sympathetic and hippocampal neurons grown in vitro, indicating that local mRNA translation also occurs in growing axonal projections [55]. Of these, the ribosomal proteins *L10A* (*RPL10A*) and *S6* (*RPS6*) were the most relevant hub genes identified in our network analysis (Figure 4A). Other translational proteins identified in neurons in vitro including the larger and smaller ribosomal subunits *RPL5*, *RPL13* and *RPS24* were also shared with our study. Impaired protein translation and ribogenesis, reflecting the importance of local translation machinery, have been extensively implicated in the pathogenesis of ALS, while it is rather novel in SOD1-ALS, respectively [7,21,35,56,57,58]. Therefore, more experimental studies investigating and clarifying the potential involvement of these ribosomal proteins on the disease progression of SOD1-ALS are necessary to confirm the results of this study.

Additionally, the GO/pathway analysis on functionally grouped modules in SOD1-ALS revealed that the upregulated genes in module 2 were mainly involved in mitochondrial complex assembly (NADH to ubiquinone) (Appendix A). Notably, the results from module analysis of the PPI network were consistent with the results of GO functional annotations of these DEGs. Interestingly, there have been several transcriptome investigations in SOD1 human samples, motor-neuron like NSC34 culture and many animal models [30,59,60,61,62]. These studies have reported dysregulation of genes involved in pathways related to oxidative stress, mitochondria, fatty acid/lipid metabolism, synapse and neurodevelopment, respectively. In accordance with these findings, we also observed an upregulation of complex I activity as well as increased ROS levels. It is worth mentioning that upregulation of mitochondrial genome encoded complex I subunits could represent a compensatory mechanism. Moreover, post-mitotic neurons are highly vulnerable to mitochondrial defects, which are pathogenically triggered by simultaneous failures of RNA processing, suppression of protein synthesis and ATP supply [63,64,65]. Intriguingly, we also uncovered that mRNAs encoding mitochondrial ribosomes were selectively downregulated in SOD1-ALS, suggesting potential deficits in mitochondrial translation as well (Figure 3A). Overall, our findings from the enrichment analysis support protein translation as an important function for understanding the pathogenesis of SOD1-ALS; however, the exact mechanism is unclear, and further biochemical experiments are needed to validate these results in more detail.

In TDP43-ALS, the expression of several genes is dysregulated affecting up to a third of the transcriptome [27,66,67], leading to the defect of multiple biological processes including alterations in RNA metabolism, mitochondrial dysfunction/oxidative stress, altered protein transport, apoptosis, and DNA damage/genomic instability. Through PPI network analysis of DEGs in TDP43, we also identified key genes and significant modules of the interaction networks enriched in transcription, splicing, and DNA repair molecular pathways. Implications of TDP43 in multiple steps of RNA metabolism such as transcription, splicing, RNA stability, microRNA processing, and mRNA transport were well established in many model systems [5,25,26]. TDP43 is also known to act as a splicing regulator whose depletion or overexpression can affect the alternative splicing of several genes [27,66] and expression of these genes was reported to be dysregulated in human CNS tissues from TDP43 ALS cases [68,69]. An important regulatory machinery during RNA splicing in eukaryotes is the spliceosome, composed of small nuclear ribonucleoproteins (snRNAs) including U1, U2, U4, U5, and U6 snRNA, in addition to a range of small nuclear RNAs (snRNPs) [70]. Reports showed that the expression profiles of such snRNAs were altered in TDP-43-knocked down cells and spinal cord from ALS patients [71,72]. Interestingly, our enrichment analysis of the two modules have revealed large amount of upregulated U6 snRNA or spliceosome assembly, and RNA polymerase II transcription machinery genes, highlighting that abnormal accumulations of TDP-43 in MNs could be responsible for this outcome (Figure 3D,F,H and Appendix A). Since a previous study demonstrated that upon TDP43 depletion the expression level of the U6 snRNA was significantly decreased [73], we hypothesize that the effect in our TDP43 mutant motor neuron is due to toxic gain of function of TDP43 resulting in neuronal death. Other important functions such as regulation of nuclear division, genome integrity, and DNA replication pathways were also identified in TDP43-ALS datasets. Indeed, patients’ motor neurons derived from ALS-linked mutations showed upregulation of the p53 pathway, high levels of DNA damage during long term culture, followed by ROS production [74,75,76]. Moreover, loss of TDP43 was associated with DNA damage and compromised cell viability in patient motor neurons [77,78,79]. Notably, our analysis indicated that the expression levels of the DNA repair genes were increased in TDP43 patients compared with healthy controls, suggesting that there might be a molecular link between the increased DNA repair machinery observed upon TDP43 mutation. Additionally, we also identified misregulation of different genes (e.g., *KIF14*) that are involved in microtubule-based transport pathways and disease progression, consistent with the literature findings showing that TDP43 may physically interfere with mitochondria and impair their transport in motor neurons [6,47,80]. Collectively, our PPI analysis uncovers many key functions of TDP-43 including transcriptional regulation, and maintenance of genomic integrity by recruitment of DNA repair factors, thus providing new insights into the pathogenesis of TDP-43-ALS. Future experimental validation should be aimed at identifying conditions as well as biological processes in which these hub genes and pathways are involved so that their potential therapeutic properties can be monitored and analyzed.

Finally, a gene regulatory network containing hub genes–TFs was constructed to better understand the process of gene regulation. Upon analysis, peroxisome proliferator activated receptor gamma (PPARG), GATA-binding factor 2 (GATA2), and forkhead box C1 (FOXC1) were predicted as the most significant TFs in both SOD1- and TDP43-ALS showing their direct interactions with target genes associated with ribosome and transcription functions (Figure 4, Table 1). The involvement of PPARG in mitochondrial biogenesis and cell survival in neurons has been extensively implicated in neurodegenerative diseases [81,82]. Activation of PPARG has been shown to confer neuroprotection in Drosophila models overexpressing TDP43 or FUS, and in SOD1 mouse models [83,84,85]. In addition, network analysis identified important TFsFOXC1 and GATA2 regulating transcriptomic changes in Alzheimer’s disease and neurodegeneration [86,87]. Thus, associations between the identified hub genes and TFs demonstrated that they may influence important toxicity pathways in these two ALS models, but experimental studies are crucial to understand their implications in the two ALS models. Taken together, the transcriptome analyses presented here extend the basis for a better understanding of the underlying molecular mechanisms of SOD1- and TDP43-ALS. Our results further suggest that targeting gene-specific alterations may be the best strategy when investigating new therapeutic approaches for genetic ALS and other neurodegenerative diseases.

Regardless of the aforementioned strengths, there are still some limitations in our study. Firstly, our sample size is relatively small and, results are purely in silico based. Hence our analyses are not free from potential random errors and false positive DEGs. Secondly and more importantly, the results of the current study are entirely based on bioinformatics prediction without subsequent experimental validation. However, for the PPI modelling, we used databases which included only experimentally proven protein-protein interactions. Thus, future studies are needed to experimentally prove our findings. Finally, although we intended to have uniform age and gender-matched controls in both the SOD1 and TDP43 datasets, we suggest that the disease lines were quite different and in fact the controls were well age- and sex-matched in case of SOD1, but not perfectly in TDP43. However, we wanted to use the same controls for both cases to avoid changes due to difference in controls.

## 4. Materials and Methods

### 4.1. Patient Characteristics

We included patient cell lines carrying mutations in *SOD1* (female, age at biopsy 46, D90A [one clone], male, age at biopsy 59, R115G [one clone]) and in *TDP43* (a “benign” S393L, late onset primary anarthria with ALS/LMND, no clinical symptoms of FTD, female, age at biopsy 85, family history of ALS and PD and a “malign” G294V, early onset ALS, no clinical symptoms of FTD, male, age at biopsy 46, no family history, two clones from patient), which were identified by Sanger sequencing in clinical settings prior to fibroblast derivation and were compared to three different wildtype cell lines from healthy volunteers (female, age at biopsy 53; male, age at biopsy 60; and female, age at biopsy 45). An overview of the used cell lines is given in Table 1. All experiments were in accordance with the Helsinki convention and approved by the Ethical Committee of the Technische Universität Dresden (EK45022009, EK393122012) and patients and healthy volunteers gave their written consent prior to skin biopsy.

### 4.2. Generation and Expansion of Cell Lines

Fibroblast cell lines were established from skin biopsies obtained from familial ALS patients and healthy controls. In case of the control lines, genetic testing was performed and they were only included if this was negative for mutations in the four main ALS genes *C9ORF72, SOD1, FUS and TARDBP*. The reprogramming procedure to obtain iPSC from fibroblasts and characterization of control iPSC lines was described previously [44,45,47].

The generation of human neural progenitor cells (NPC) and motor neurons (MN) was performed as previously reported [45] based on the protocol by Reinhardt et al. [43,88]. Importantly, the NPC culture was a resource for final MN differentiation, which was initiated by treatment with 1 µM purmorphamine (PMA) in N2B27 and supplemented with 1 µM retinoic acid (RA) on the third day. To increase the purity of MN enriched cell culture another split was performed on day 9 of the protocol. In parallel, the medium constitution was changed: Instead of PMA and RA, 10 ng/µL BDNF, 500 µM dbcAMP and 10 ng/µL GDNF was added to N2B27 ensuring neuronal maturation. Motor neurons were harvested after being cultured for 30 days [42,46,47].

### 4.3. RNA Isolation and Transcriptome Analysis

After 30 DIV, each culture was washed by gently replacing the maturation medium with PBS warmed to 37 °C. Total RNA from ~5 × 10^6^ cells was extracted using the RNeasy Kit (QIAGEN) according to the manufacturer’s protocol including a column DNase digest. The RNA was eluted in RNase-free water and RNA quality was assessed by measuring the ratio of absorbance at 260/280 nm using a Nanodrop 1000 Spectrometer (Thermo Scientific, Waltham, MA, USA).

For RNA-Seq, motor neurons of the cell lines wildtype controls, SOD1-ALS and TDP43-ALS were generated in biological triplicates (3 independent differentiations side by side) and 1 µg of total RNA was isolated and validated as previously described and was used for selection of Poly(A) plus RNA and library preparation was done after oligo (dT) selection. RNA-Seq libraries were generated by performing RNA fragmentation, cDNA synthesis, linker ligation and PCR enrichment. These libraries were then subjected to paired-end sequencing on an Illumina HiSeq2500 platform (Illumina, San Diego, CA, USA) to obtain reads. The raw reads were quality controlled (QC) and verified before mapping. A total of 180 million pairs of RNA reads of size 100 bp were generated from the 8 samples and were uniquely aligned to the human reference genome (hg38 (obtained from Ensembl assembly v100) after a cleaning step to remove low quality regions. After trimming adaptor sequences and removing low quality reads, we got 6 to 7 million reads per sample. The GC content was 30% and the percentage of reads with ≥Q30 were 99% in all samples. On average, ~92% of total high quality clean reads were aligned to the reference human genome hg38_ensembl_release100.The alignment was performed using STAR aligner program with default parameters. For further preprocessing and quality control, aligned sequence reads (BAM files) were uploaded into Partek Genomics Suite version 7.0 software (Partek Inc., St Louis, MO, USA) using standard pipeline settings provided by Partek software. Raw read counts were obtained by quantitating aligned reads with annotated human genes hg38_Ensembl v100, using the Expectation Maximization (EM) algorithm, in which isoform expression levels are quantified across the whole genome at the same time [89]. Details of the Partek EM algorithm can be found in the White Paper on RNA-Seq Methods [90]. Normalization of the raw read counts was performed with the Partek software using Reads Per Kilobase per Million (RPKM) normalization method [91]. The differential gene expression analysis was performed using Partek software’s ANOVA model statistical approach with default parameters to analyze the difference between wildtype control and ALS mutant cell lines. To generate significant DEGs among different samples, a cutoff of the false discovery rate *p*-value ≤ 0.05 (Benjamini–Hochberg correction) and│log_2_FC**│** ≥ 1.5 were applied. The lists of DEGs were used for gene ontology and PPI analysis using Metascape, EnrichR, DAVID, STRING and NetworkAnalyst databases.

### 4.4. Functional Annotation of DEGs by GO and KEGG Analysis

To discern the implications of DEGs in SOD1 and TDP43-ALS, we performed functional and pathway enrichment analyses of the mapped genes using the most commonly used web-based online platforms such as Metascape [92], EnrichR [93] and DAVID [94]. Gene ontology (GO) analysis is a common method in functional enrichment analysis, aiming to provide biological attributes of DEGs such as biological processes (BP), molecular functions (MF), and cellular components (CC). Kyto Encyclopedia of Genes and Genomes (KEGG) http://www.genome.jp/kegg, accessed on 20 June 2022 [95] database is crucial in understanding different pathways, their associated functions in a given biological system, and the molecular level information using the large-scale datasets generated from genome sequencing. The different databases provided similar information with the majority of the genes acting in RNA metabolic processes (Appendix A and Appendix A).

Metascape (http://metascape.org/gp/index.html#/main/step1, accessed on 6 July 2022) is a powerful gene function annotation tool, which is involved in four processes: ID conversion, gene annotation for a large number of genes or proteins, enrichment analysis and construction of PPI networks [92]. Metascape integrates several functional databases, such as GO, KEGG and Uniprot to analyze multiple gene sets simultaneously. The DEGs were analyzed and visualized by Metascape with the criteria of minimum overlap >3, *p*-value ≤ 0.05, and minimum enrichment score > 1.5.

EnrichR (http://amp.pharm.mssm.edu/Enrichr/, accessed on 6 July 2022) is a web-based tool that allows the evaluation of annotations with its extensive gene set libraries [93]. The significant GO terms and KEGG (http://www.genome.jp/kegg) pathways were selected with a threshold *p*-value ≤ 0.05.

DAVID (The Database for Annotation, Visualization, and Integrated Discovery) (Version 6.8 https://david.ncifcrf.gov/home.jsp, accessed on 6 July 2022) [94] is an online tool allowing a comprehensive analysis of a large list of DEGs to identify enriched biological GO terms, and visualize genes on KEGG pathway maps. Enrichment analysis was performed with a *p*-value significance level of ≤0.05 and a gene count ≥3.

### 4.5. PPI Network Construction and Identification of Hub Genes

To better understand the functional interactions of the DEGs and identify the most important candidate genes in ALS disease subtypes, a comprehensive protein-protein interaction (PPI) network of their encoding products was constructed by using the STRING (The Search Tool for the Retrieval of Interacting Genes/Proteins) database (Version 11.0, https://string-db.org/, accessed on 9 July 2022) [48]. The STRING database collects and integrates all functional associations between the genes/proteins by consolidating known and predicted interaction data derived from sources including databases, high-throughput experiments, co-expression, text mining, neighborhood and gene fusion with the highest confidence score. The statistical enrichment analyses in STRING indicated that the DEGs were significantly enriched in PPI networks (*p*-value ≤ 0.05). Based on experimentally-derived interactions, with a high confidence score of 0.7 (confidence score ≥ 0.7), and a maximum number of interactions to top 20, two PPI networks were generated by mapping total DEGs for each SOD1- and TDP43-ALS subtypes respectively. Subsequently, PPI networks were analyzed by Cytoscape (version 3.8.2, http://cytoscape.org/, accessed on 28 June 2022) software [49], an open source software for visualizing complex biomolecular interaction networks containing diverse plugins for multiple analyses. Cytoscape represents PPI networks as graphs with nodes illustrating proteins and edges representing associated interactions. The significant hub nodes in the PPI network were selected according to the scoring of maximum correlation criterion (MCC) by using the Cytoscape plugin cytoHubba (https://apps.cytoscape.org/apps/cytohubba, accessed on 28 June 2022), [53] which explores important nodes and modules by topological algorithms. The topological parameter indicates the importance of a node (protein) as functionally connecting link in a PPI network, suggesting an important biological function. The top ten genes scoring the highest in the PPI network were identified as hub genes in our present study.

### 4.6. Module Identification and Enrichment Analysis

The Cytoscape plugin Molecular Complex Detection (MCODE, version 2.0.0; http://apps.cytoscape.org/apps/mcode, accessed on 28 June 2022) [50] was applied to extract highly interconnected modules that detect densely connected regions in large PPI networks that may represent molecular complexes, with the default analysis criteria as degree cut-off = 2, node score cut-off = 0.2, K-core = 2 and maximum depth = 100 [50]. Significant modules were identified with MCODE score ≥ 5 and nodes ≥ 5.

The functional enrichment analyses of these predicted genes was performed by ClueGO/CluePedia plugin of Cytoscape (version 3.8.2) in each module [51,52]. ClueGo integrates gene ontology (GO) terms and enriched KEGG pathways and creates a functionally organized pathway term network. *p*-values ≤ 0.05 were considered to be significant. Validation of molecular/biological function of ALS subtypes and finding potentially essential genes can be inferred through these analytical results.

### 4.7. Gene-Transcription Factor Interaction Analysis

Official gene identifiers from the dysregulated hub genes were imported into NetworkAnalyst tool (http://www.networkanalyst.ca/faces/home.xhtml, accessed on 20 June 2022) [96] for network analysis of transcription factors (TFs) and JASPER database was used to identify TF-gene interactions. NetworkAnalyst is a comprehensive web-based platform for network-based visual analytics of gene expression profiling, metaanalysis, and biological interpretation. The JASPER database includes curated, non-redundant transcription factor-binding profiles based on experimental methods [97]. The network topological parameters (i.e., degree and betweenness centrality) were used to rank the identified transcription factors that regulated the expression of hub genes. A hub gene-TF regulatory network was constructed and analyzed in NetworkAnalyst [96].

### 4.8. Statistical Analysis

RNA seq data were statistically analyzed using Partek Genomics Suite software (Partek Inc., Chesterfield, MO, USA). All of the gene expression samples presented in this study were designed for 3 biological replicates (mean ± SD, *n* ≥ 3). *p*-value *≤* 0.05 and log_2_FC ≥ 1.5 or log_2_FC ≤ −1.5 were considered as significant thresholds for the identification of DEGs. For the functional enrichment and PPI/GGI/regulatory network analysis, significantly enriched GO terms, pathways and modules were identified using a *p*-value ≤ 0.05 as the cut off value for statistical significance.

### 4.9. Data Availability

Raw and processed data in this study were deposited in the NCBI Gene Expression Omnibus (GEO, http://www.ncbi.nlm.nih.gov/geo, accessed on 17 August 2022) with the following accession number: GSE210969.

## 5. Conclusions

The strength of the current study is that we performed integrative bioinformatics analyses of large-scale RNA-Seq expression profiling data sets to identify key genes and multiple pathways dysregulated in iPSC-derived spinal motor neurons from SOD1- and TDP43-ALS patients. By utilizing both comprehensive enrichment and PPI/regulatory network interaction analyses, our data have revealed that—on the one hand—the SOD1 transcriptome is remarkably distinct from that of TDP43. On the other hand, advanced bioinformatics condensed the multiple dysregulated pathways to two mainly affected in each condition. By this means, we identified *RPL3*, *RPS24*, *MRPL28*, *TAF5*, *LSM7*, *MT-ND6*, *NDUFA13* and *NDUFC1* as hub genes in SOD1- and TDP43-ALS patient-derived motor neurons, most of which were required for RNA metabolism and oxidative energy production. Of note, we found dysregulation of gene expression as the common denominator in both disease conditions. While this was on the level of the transcriptional machinery and splicing regulation in TDP43-ALS, we identified impaired expression of components of the protein translation/ribosome machinery in SOD1-ALS. Our findings need further experimental verification, but may provide potential useful evidence and ideas for further exploration of the underlying mechanisms of ALS pathogenesis.

## Figures and Tables

**Figure 1 ijms-23-09652-f001:**
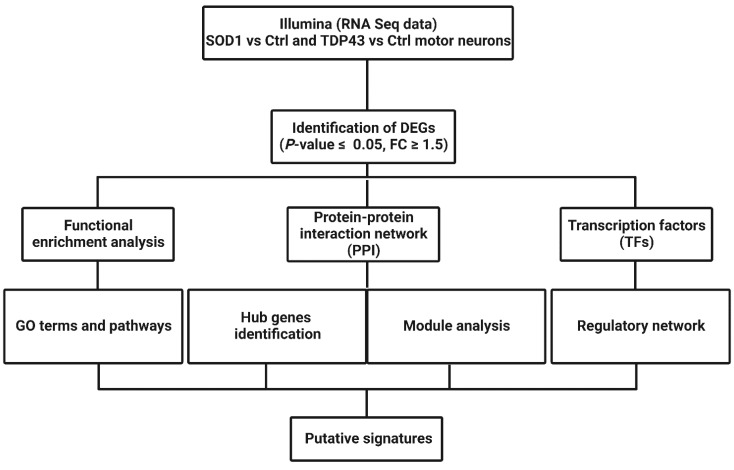
Overall workflow of the study.

**Figure 2 ijms-23-09652-f002:**
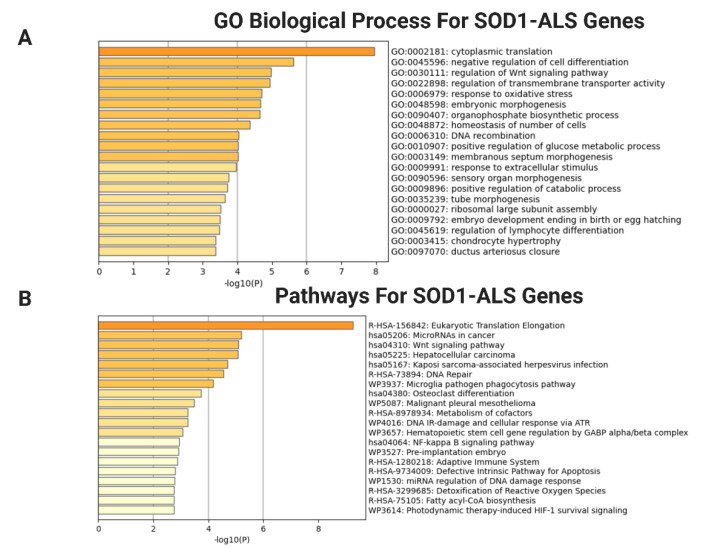
Functional enrichment analysis of DEGs by Metascape (https://metascape.org/gp/index.html#/main/step1, accessed on 6 July 2022) revealed distinct pathways in SOD1- and TDP43-ALS. (**A**) Metascape bar chart of enriched GO and (**B**) Pathway terms (KEGG/Reactome/WikiPathway) across the input DEGs in SOD1-ALS. (**C**) Metascape bar graph of GO terms and (**D**) KEGG pathways that are significantly enriched in DEGs specific for TDP43-ALS datasets. X-axis represents the statistical significance of the enrichment (−log10 (*p*-value)). Bar chart of GO and Pathway terms colored by *p*-values ≤ 0.05 were adapted from Metascape, where terms containing more genes tend to have more significant *p*-values. The length of the bar represents the significance of that specific gene-set or term. The brighter the color, the more significant that term is.

**Figure 3 ijms-23-09652-f003:**
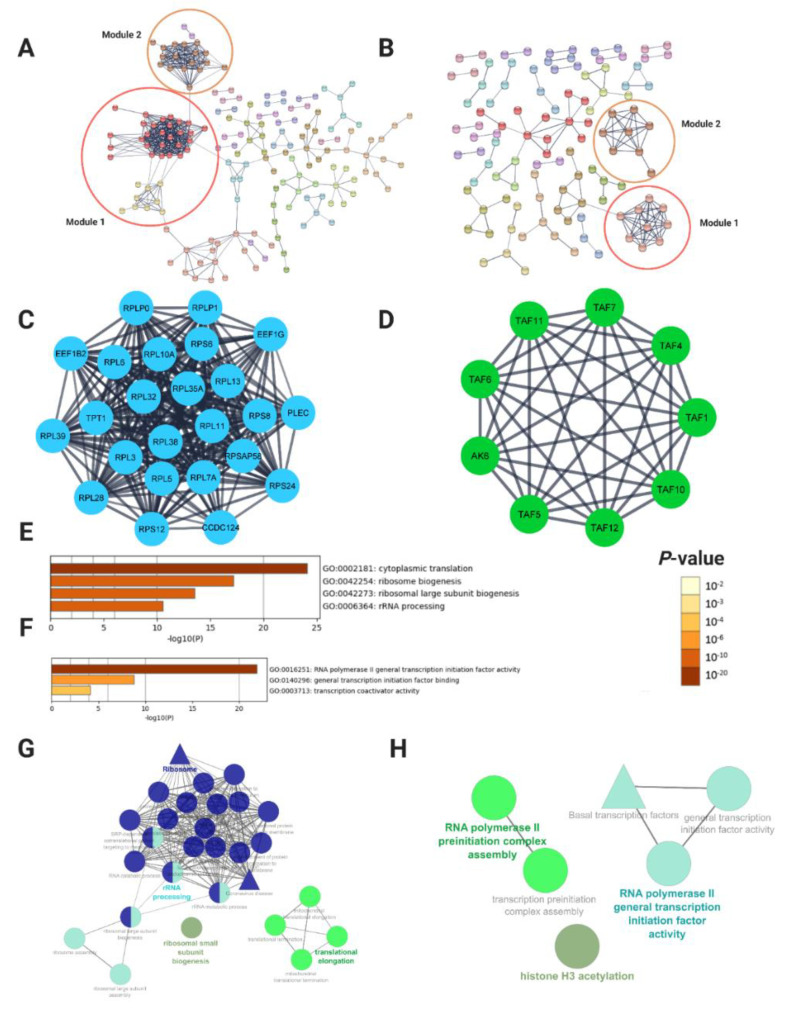
PPI network analysis of total DEGs converge to two main affected pathways in both, SOD1- and TDP43-ALS. (**A**) PPI network of SOD1- and (**B**) TDP43-ALS DEGs, in which thicker lines indicates stronger data support. The nodes indicate the DEGs and the edges indicate the interaction (experimental evidence only) between two proteins. The STRING database was used to establish functional associations among the known and predicted proteins using annotated DEGs as query for SOD1- and TDP43-ALS interaction networks, with high confidence score of >0.7 and a maximum number of interactions to top 20. (**C**) The module 1 from SOD1- and (**D**) TDP43-ALS identified from the whole PPI network. (**E**) Enrichment analysis of the module 1 from SOD1- and (**F**) TDP43-ALS by Metascape. X-axis represents the statistical significance of the enrichment (−log10 (*p*-value)). Bar chart of GO and Pathway terms colored by *p*-values were adapted from Metascape, where terms containing more genes tend to have more significant *p*-values. The length of the bar represents the significance of that specific gene-set or term. The brighter the color, the more significant that term is (**G**) Comprehensive enrichment analysis of the functional group network (only significantly enriched GO terms/Pathways are visualized, *p*-value ≤ 0.05) for module 1 in SOD1- and (**H**) TDP43-ALS using ClueGo/CluePedia plugin in Cytoscape. ClueGO/CluePedia annotation results are based on biological process (circular node) and KEGG pathway (triangle shape) analysis.

**Figure 4 ijms-23-09652-f004:**
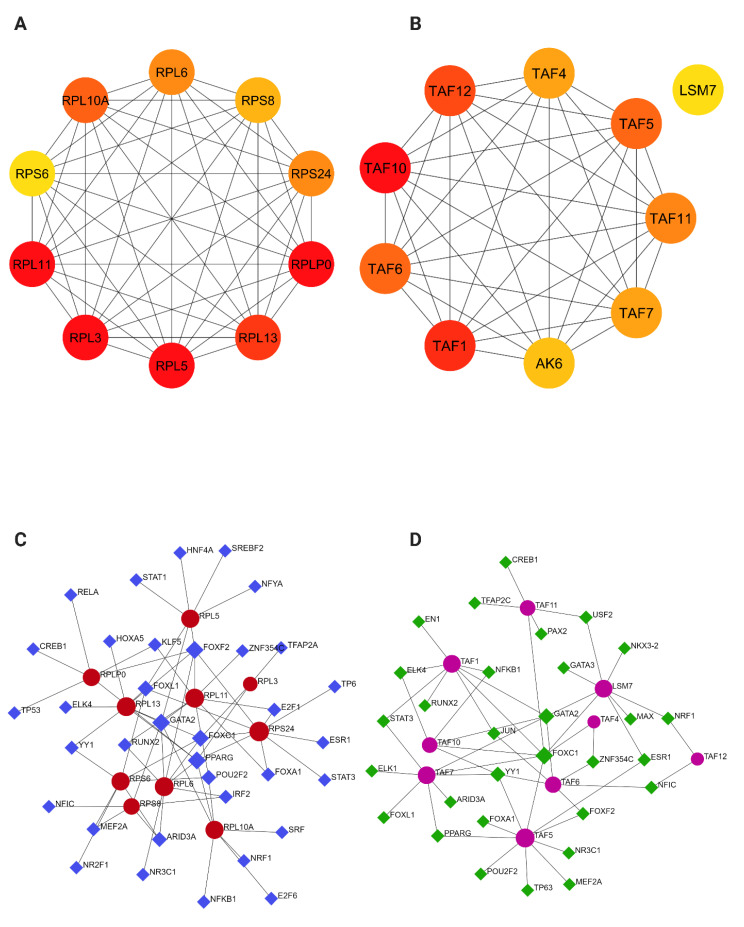
Integrated analysis on hub genes further highlighted one main pathway in SOD1- and TDP43 ALS. (**A**) Top 10 hub genes from SOD1- and (**B**) TDP43-ALS identified using the cytoHubba plugin (Cytoscape) among TDP43-ALS and SOD1-ALS datasets. The deeper the color of the node, the higher the level of significance at the PPI network. (**C**) The hub gene-transcription factor (TF) regulatory interaction networks in SOD1- and (**D**) TDP43-ALS datasets. Red and pink nodes stand for the hub gene and blue and green diamond stands for the transcription factor.

**Table 1 ijms-23-09652-t001:** Patient/proband characteristics.

Genotype	Sex	Age at Biopsy (Years)	Mutation	Family History	Age of Disease Onset	ALS Type	Clinical Characteristics	Disease Duration (Months)	Clones	DIV
Controls										
	Female	53	-	-	-	-		-	1	30
	Male	60	-	-	-	-		-	1	30
	Female	45	-	-	-	-		-	1	30
TDP43-ALS										
	Female	85	p.S393L	Pos. for ALS	85	Bulbar	Progressive anarthria, LMND, no clinical symptoms of FTD	48	1	30
	Male	46	p.G294V	neg for ALS	37	Spinal	Early onset ALS (37 years), monomelic right leg amyotrophy, no clinical symptoms of FTD	>120 (alive)	2	30
SOD1-ALS										
	Male	59	p.R115G	Pos. for ALS (Mother and Brother)	n.d	Spinal	n.d	n.d	1	30
	Female	46	p.D90A	Pos. for ALS (Brother)	41	Spinal	Slowly progressive classical spinal ALS, no cognitive impairment	204	1	30

n.d: no data; LMND: lower motor neuron disease; DIV: days in vitro.

**Table 2 ijms-23-09652-t002:** Transcription factors (TFs) of hub genes.

Cellular Model	TFs	Target Genes	Count
SOD1-ALS	PPARG	*RPL6*, *RPLP0*, *RPL3*, *RPS24*, *RPL13*, *RPL10A*	6
	GATA2	*RPL11*, *RPS8*, *RPL6*, *RPL13*, *RPL10A*, *RPL5*	6
	FOXF2	*RPL11*, *RPL6*, *RPLP0*, *RPS24*, *RPL13*, *RPL5*	6
	FOXC1	*RPS6*, *RPL6*, *RPL3*, *RPS24*, *RPL13*	5
TDP43-ALS	FOXC1	*TAF4*, *TAF7*, *LSM7*, *TAF1*, *TAF10*, *TAF11*, *TAF5*	7
	GATA2	*TAF6*, *TAF7*, *LSM7*, *TAF1*, *TAF10*	5

## Data Availability

Not applicable.

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
