# Peer review of "Downstream Effects of Mutations in *SOD1* and *TARDBP* Converge on Gene Expression Impairment in Patient-Derived Motor Neurons"

_ijms, 2022, doi:10.3390/ijms23179652_

Round 1
Reviewer 1 Report
Dash et al have carried out a preliminary transcriptome analysis of ipsc-lines, two that contain TARBP mutations and two with SOD1 mutations. This is interesting however, some sentences really overstate the size and conclusions of the study. The sample size is small and the data is multidimensional. This means it is prone to false discoveries. The study does not include any replication data but I understand there are some corroborations in the literature.
Minor updates are suggested below - to improved readability/remove errors.
Results
In text suggest being more specific about how long the each ipsc line was differentiated for, how many ipsc lines were examined, and what phenotypes were demonstrated.
line 102- table 1 does not contain details of cell lines
Figure 1 should have ipsc line/diff details.
Results- DEG only mentions P<=0.05 but the figure 1 says 1.5 fold change, p-value. Did both need to occur?
Figure 2. increase text size. Legend typo is missing a bracket near log10P. Not clear what the x-axis lines mean? Not clear what the different orange colours mean.
Figure 3. text is too small - unreadable.
Methods
line 343. not clear what a benign and malign means? Shouldn't both these lines have a highly penetrant causal mutation in TARDBP?
line 344. were the controls age-matched? sex-matched?
Table 2. Capitilisation should be consistent
line 379. million read 'per' sample.
line 383. how were the data normalised?
best practice would include at least a sex check.
line 474. data availability/accession number is missing
Throughout - gene names should be italicised
Author Response
Reviewer 1
Dash et al have carried out a preliminary transcriptome analysis of ipsc-lines, two that contain TARBP mutations and two with SOD1 mutations. This is interesting however, some sentences really overstate the size and conclusions of the study. The sample size is small and the data is multidimensional. This means it is prone to false discoveries. The study does not include any replication data but I understand there are some corroborations in the literature.
Response: We deeply thank the reviewer for his overall positive evaluation. We carefully revised the manuscript and tried hard to not overstate our results. But please be aware that we included 3 TDP43 lines and 2 SOD1 lines, but these were derived from two unrelated donors, respectively.
Minor updates are suggested below - to improved readability/remove errors.
Results
In text suggest being more specific about how long the each ipsc line was differentiated for, how many ipsc lines were examined, and what phenotypes were demonstrated.
Response: We include this information now also in the results section, besides keeping this also in the material and methods section as before
line 102- table 1 does not contain details of cell lines
Figure 1 should have ipsc line/diff details.
Response: We appreciate the reviewer’s comments. However, since these lines have been already published a couple of times having exact this information about differentiation readouts and quantifications, we would rather like to cross ref instead of redundantly present such a quantification
Results- DEG only mentions P<=0.05 but the figure 1 says 1.5 fold change, p-value. Did both need to occur?
Response: Please do apologize for being not clear enough. The reviewer is correct, both needed to occur. We now clarify this in the revised version of the manuscript.
Figure 2. increase text size. Legend typo is missing a bracket near log10P. Not clear what the x-axis lines mean? Not clear what the different orange colours mean.
Response: We thoroughly revised both the figure and the figure legend.
Figure 3. text is too small - unreadable.
Response: We thoroughly revised both the figure and the figure legend.
Methods
line 343. not clear what a benign and malign means? Shouldn't both these lines have a highly penetrant causal mutation in TARDBP?
Response: We apologize for being not clear enough. We wanted to state that the patients from which the skin biopsy was taken had significant different time of onset and clinical disease course. We now redraw this paragraph and added this information in the revised table 1
line 344. were the controls age-matched? sex-matched?
Response: We thank the reviewer for this question. Indeed, our we intended to have a uniform age- and gender matching for the SOD1 and TDP43 datasets. Unfortunately, the disease lines were quite different, that in fact the controls were well age- and sex-matched in case of SOD1, but not perfectly for TDP43. But we wanted to use the same controls for both cases to avoid changes due to differences in controls. We state this however much clearer in the revised version of the manuscript including a statement as potential limitation of the study
Table 2. Capitalisation should be consistent
line 379. million read 'per' sample
line 383. how were the data normalised?
Response: We add this information in more details
best practice would include at least a sex check
line 474. data availability/accession number is missing
Response: added
Throughout - gene names should be italicised
Response: Gene names were italicised, but protein names not

Reviewer 2 Report
In the manuscript "Downstream effects of mutations in SOD1 and TARDBP converge on protein translation impairment in patient-derived motor neurons" by Banaja P Dash et al., the authors use human induced pluripotent stem cell-derived motor neurons harbouring SOD1 and TDP43 mutations to conduct comprehensive high-throughput RNA-sequencing (RNA-Seq) analysis and study mRNA expression profiles that could be relevant to understand ALS pathogenesis harbouring different types of mutations. The theme is interesting and extremely relevant for the field. However, there are several aspects that merit the attention of the authors, which are the following:
1. One important aspect is the global organization of the paper. The material and methods section is placed between the introduction and conclusion. The material and methods section should come after the introduction, so that the readers understand the methods employed and then better judge the results obtained, as well as, the discussion and conclusion sections.
2. The material and methods section needs considerable improvements, specially when it comes to the patient details and to the exact protocols employed to differentiate the motor neurons from human induced pluripotent stem cells. Are the patients dead or alive? How many months of disease after the initial symptoms? Furthermore, what is the purity of the MN cultures from which the RNA studies were conducted? Previous studies highlighted the relevance of using pure cultures of hESC-MNs and hiPSC-MNs when studying human motor neurons in vitro (please view Lamas NJ, Johnson-Kerner B, Roybon L, Kim YA, Garcia-Diaz A, Wichterle H, Henderson CE. Neurotrophic requirements of human motor neurons defined using amplified and purified stem cell-derived cultures. PLoS One, 9(10): 1-13, 2014.). How many cells were employed in each RNA sequencing study? How many replicates per patient were used? The description of all these aspects will help to critically improve the quality of the paper.
2. The introduction is well-organized, but it would benefit from expanding the description of MN-centered mechanisms that are responsible for MN death in ALS. In addition, there is no single mention to the non-motor neuron centered mechanisms that are responsible for MN death in ALS. For example, it would be fundamental to briefly add information on the neurotrophic support and also on astrocyte toxicity in ALS. Please view and cite the following recent review papers:
- Brown, R. H., and Al-Chalabi, A. (2017). Amyotrophic Lateral Sclerosis. N. Engl. J. Med. 377 (2), 162–172. doi:10.1056/NEJMra1603471.
- Lamas NJ, Roybon L. Harnessing the Potential of Human Pluripotent Stem Cell-Derived Motor Neurons for Drug Discovery in Amyotrophic Lateral Sclerosis: From the Clinic to the Laboratory and Back to the Patient. Frontiers in Drug Discovery, 1: 1-26, 2021.
3. The D90A SOD1 mutation studied is a ALS SOD1 variant that allows patients to be only mildly affected by the disease, with patient survival usually longer than 10 years [please views Andersen, P. M., Forsgren, L., Binzer, M., Nilsson, P., Ala-Hurula, V., Keränen, M. L., et al. (1996). Autosomal Recessive Adult-Onset Amyotrophic Lateral Sclerosis Associated with Homozygosity for Asp90Ala CuZn-Superoxide Dismutase Mutation. A Clinical and Genealogical Study of 36 Patients. Brain 119, 1153–1172. doi:10.1093/brain/119.4.1153]. It would be important to discuss this aspect to further highlight the relevance of the results obtained.
4. Importantly, the authors ackowledged the limitations of their study in the second paragraph of the conclusion. However, that valuable information would be better placed in the discussion section.
5. The Data Availability section lacks the accession number (Raw and processed data in this study were deposited in the NCBI Gene Expression 473 Omnibus (GEO, http://www.ncbi.nlm.nih.gov/geo) with the following accession number:).
Therefore, the present manuscript is interesting and relevant, but it will be mandatory to perform extensive changes before if it is ready to be published in the International Journal of Molecular Sciences.
Author Response
Reviewer 2
In the manuscript "Downstream effects of mutations in SOD1 and TARDBP converge on protein translation impairment in patient-derived motor neurons" by Banaja P Dash et al., the authors use human induced pluripotent stem cell-derived motor neurons harbouring SOD1 and TDP43 mutations to conduct comprehensive high-throughput RNA-sequencing (RNA-Seq) analysis and study mRNA expression profiles that could be relevant to understand ALS pathogenesis harbouring different types of mutations. The theme is interesting and extremely relevant for the field. However, there are several aspects that merit the attention of the authors, which are the following:
- One important aspect is the global organization of the paper. The material and methods section is placed between the introduction and conclusion. The material and methods section should come after the introduction, so that the readers understand the methods employed and then better judge the results obtained, as well as, the discussion and conclusion sections.
Response: We do fully agree with the reviewer and would love to present the material and methods section after the introduction; however, the journal’s policy is different and the current representation is according to the journal’s template. Thus, we are unable to address this except the editorial office will allow to present our paper different to their usual format.
- The material and methods section need considerable improvements, especially when it comes to the patient details and to the exact protocols employed to differentiate the motor neurons from human induced pluripotent stem cells. Are the patients dead or alive? How many months of disease after the initial symptoms?
Response: We included this information in the revised table
Furthermore, what is the purity of the MN cultures from which the RNA studies were conducted? Previous studies highlighted the relevance of using pure cultures of hESC-MNs and hiPSC-MNs when studying human motor neurons in vitro (please view Lamas NJ, Johnson-Kerner B, Roybon L, Kim YA, Garcia-Diaz A, Wichterle H, Henderson CE. Neurotrophic requirements of human motor neurons defined using amplified and purified stem cell-derived cultures. PLoS One, 9(10): 1-13, 2014.). How many cells were employed in each RNA sequencing study? How many replicates per patient were used? The description of all these aspects will help to critically improve the quality of the paper.
Response: We really appreciate the reviewer’s comments. We did publish the respective cell lines more than once including thorough description of differentiation marker expression, quantification of motor neurons, neurotransmitter identity and electrophysiological properties. Nevertheless, we revised the results section to not only cross ref to the respective papers but also reporting more details on this at the beginning of the manuscript.
- The introduction is well-organized, but it would benefit from expanding the description of MN-centered mechanisms that are responsible for MN death in ALS. In addition, there is no single mention to the non-motor neuron centered mechanisms that are responsible for MN death in ALS. For example, it would be fundamental to briefly add information on the neurotrophic support and also on astrocyte toxicity in ALS. Please view and cite the following recent review papers:
- Brown, R. H., and Al-Chalabi, A. (2017). Amyotrophic Lateral Sclerosis. N. Engl. J. Med. 377 (2), 162–172. doi:10.1056/NEJMra1603471.
- Lamas NJ, Roybon L. Harnessing the Potential of Human Pluripotent Stem Cell-Derived Motor Neurons for Drug Discovery in Amyotrophic Lateral Sclerosis: From the Clinic to the Laboratory and Back to the Patient. Frontiers in Drug Discovery, 1: 1-26, 2021.
Response: We agree with the reviewer and briefly state the role on neurotrophic support and astrocytes in ALS pathophysiology. But we still kept this short, since our differentiation protocol yields neuronal cultures without astrocytes, thus our transcription data is not influenced by astrocyte or other neuroectodermal cell influences
- The D90A SOD1 mutation studied is a ALS SOD1 variant that allows patients to be only mildly affected by the disease, with patient survival usually longer than 10 years [please views Andersen, P. M., Forsgren, L., Binzer, M., Nilsson, P., Ala-Hurula, V., Keränen, M. L., et al. (1996). Autosomal Recessive Adult-Onset Amyotrophic Lateral Sclerosis Associated with Homozygosity for Asp90Ala CuZn-Superoxide Dismutase Mutation. A Clinical and Genealogical Study of 36 Patients. Brain 119, 1153–1172. doi:10.1093/brain/119.4.1153]. It would be important to discuss this aspect to further highlight the relevance of the results obtained.
Response: On the one hand, we do agree with the reviewer. However, different gene mutations can cause significantly different disease courses and phenotypes. This is especially true in case of SOD1, yielding also to different mitochondrial and aggregation phenotypes (see Günther et al, Cells 2022). Since we wanted to look for overarching phenotypes, we intentionally included also a D90A cell line. We made this however clearer in the revised version of the manuscript
- Importantly, the authors acknowledged the limitations of their study in the second paragraph of the conclusion. However, that valuable information would be better placed in the discussion section.
Response: We did so.
- The Data Availability section lacks the accession number (Raw and processed data in this study were deposited in the NCBI Gene Expression 473 Omnibus (GEO, http://www.ncbi.nlm.nih.gov/geo) with the following accession number:).
Response: added
Therefore, the present manuscript is interesting and relevant, but it will be mandatory to perform extensive changes before if it is ready to be published in the International Journal of Molecular Sciences.
Response: we deeply appreciate the very positive overall statement. As the reviewer can see, we tried hard to address every single comment raised by both authors. We believe that the quality of our manuscript thereby improved significantly and deeply hope that it is now acceptable for publication.

Round 2
Reviewer 2 Report
In the revised version of the original manuscript "Downstream effects of mutations in SOD1 and TARDBP converge on protein translation impairment in patient-derived motor neurons" by Banaja P Dash et al., the authors addressed well all the points raised in the initial version of the manuscript. Therefore, I strongly recommend its publication in the International Journal of Molecular Sciences.